# *Cannabis sativa* CBD Extract Exhibits Synergy with Broad-Spectrum Antibiotics against *Salmonella enterica* subsp. *Enterica* serovar *typhimurium*

**DOI:** 10.3390/microorganisms10122360

**Published:** 2022-11-29

**Authors:** Logan Gildea, Joseph Atia Ayariga, Junhuan Xu, Robert Villafane, Boakai K. Robertson, Michelle Samuel-Foo, Olufemi S. Ajayi

**Affiliations:** 1The Microbiology Program, Department of Biological Sciences, College of Science, Technology, Engineering, and Mathematics (C-STEM), Alabama State University, Montgomery, AL 36104, USA; 2The Industrial Hemp Program, Department of Biological Sciences, Alabama State University, Montgomery, AL 36104, USA

**Keywords:** *Salmonella*, novel antibacterial agents, cannabidiol, co-therapy, cell-membrane integrity

## Abstract

New generation antibiotics are needed to combat the development of resistance to antimicrobials. One of the most promising new classes of antibiotics is cannabidiol (CBD). It is a non-toxic and low-resistance chemical that can be used to treat bacterial infections. The antibacterial activity of *Cannabis sativa* L. byproducts, specifically CBD, has been of growing interest in the field of novel therapeutics. As research continues to define and characterize the antibacterial activity that CBD possesses against a wide variety of bacterial species, it is important to examine potential interactions between CBD and common therapeutics such as broad-spectrum antibiotics. In this study it is demonstrated that CBD-antibiotic (combination of CBD and antibiotic) co-therapy can effectively fight *Salmonella typhimurium* (*S. typhimurium*) via membrane integrity disruption. This research serves to examine the potential synergy between CBD and three broad-spectrum antibiotics (ampicillin, kanamycin, and polymyxin B) for potential CBD-antibiotic co-therapy. In this study, it is revealed that *S. typhimurium* growth is inhibited at very low dosages of CBD-antibiotic. This interesting finding demonstrates that CBD and CBD-antibiotic co-therapies are viable novel alternatives to combating *S. typhimurium*.

## 1. Introduction

Historically, *Cannabis sativa* L. has been utilized in a multitude of applications ranging from textile production to therapeutic treatment [1,2,3,4,5]. As we move towards the mid-21st century, public acceptance of *C. sativa*-derived products continues to increase [6]. This growth in public acceptance entices exploration of applications for *C. sativa* and its byproducts. Currently, *C. sativa* and its byproducts have been shown to be efficacious against several neurological diseases including Tourette’s syndrome, epilepsy, and multiple sclerosis [7,8,9]. The characterization of two active compounds of *C. sativa*, tetrahydrocannabinol (THC) and cannabidiol (CBD), has significantly progressed our understanding of the physiological and psychological effects of *C. sativa* on humans [10,11,12,13]. CBD, the primary focus of our research, is one of the two active compounds of *C. sativa*. CBD is a highly lipophilic molecule, making it naturally water soluble and induces interactions with other lipids. This compound has been characterized as anti-inflammatory agents with a notable ability to interact with cellular membranes [14,15]. In addition to this, CBD is a promoter of drug–drug interactions and can even potentiate the action of many drugs [16]. Research has demonstrated the efficacy of CBD against several neurological disorders including epilepsy, multiple sclerosis, and Tourette’s syndrome [7,8,9]. While CBD has been extensively researched for use in neurological disorders, there has been a growing interest in other potential applications of CBD, especially as an antimicrobial agent [17,18,19,20,21]. This increasing interest in CBD and its versatility has promoted further study into other novel potential applications of this botanical extract.

A topic of growing interest is the antibacterial activity of CBD against several pathogenic bacteria. Recent studies have observed CBD’s antibacterial activity against several clinically relevant pathogens including *Staphylococcus aureus*, *Streptococcus pneumoniae*, *Clostridium difficile*, *Neisseria* spp., *Moraxella catarrhalis*, *Legionella pneumophila*, and *Salmonella* spp. [17,18,19,20,21]. These studies have promoted further research into CBD and its antibacterial capabilities. Recent studies have confirmed that the antibacterial activity of CBD is carried out by membrane disruption through lipid-lipid interactions, however this mechanism has not been fully defined [18,22]. These studies have promoted CBD as a ‘state of the art’ botanical therapeutic with broad application against a wide variety of bacterial species. With the increased prevalence and occurrence of antimicrobial resistance posing a significant threat to public health, the development of novel antibacterial agents is a necessity [23,24,25,26]. The current rise in antibiotic and multidrug resistant bacteria is a major crisis resulting in 2.8 million infections and 35,000 deaths annually in the United States alone [27]. In addition, resistant bacteria present a major economic burden of approximately 7.7 billion dollars annually in the United States because of prolonged hospitalizations and exhaustion of treatment avenues [28]. The development of novel therapeutic agents, such as CBD, which represents a new and novel class of potential antibiotics with a unique chemical structure and low detectable resistance, is crucial in the mitigation of the crisis that resistant pathogens are responsible for [29,30,31,32,33].

*Salmonella* species are one of the most common and prevalent foodborne pathogens worldwide, found in several food products including poultry, seafood, and other fresh or processed meats [19,26,34,35,36,37]. Most notable resistant bacteria are associated with hospital-acquired infections; however, the occurrence of resistant foodborne bacteria could present an even greater threat to public health. Accounting for 1.35 million infections, 26,500 hospitalizations, and 420 deaths annually in the United States alone, *Salmonella* species have a significant impact on public health [27]. The mass and often inappropriate use of antibiotics, especially in the food industry, is a major concern and could lead to mass exposure of the public to pathogens that are resistant to one or more antibiotic treatments [19,23,24,25,26,37,38,39]. *S. typhimurium*, the strain of focus in this study, is notable for its ability to develop resistance to antibiotics. According to the Centers for Disease Control (CDC), 59% of ampicillin-resistant *Salmonella* infections were attributed to *S. typhimurium* [27]. Thus, making *S. typhimurium* a relevant bacterium for the study and development of novel antibacterial agents.

CBD’s antibacterial potential has been a topic of growing interest [18,19,20,21,29,30]. Recent studies suggest that the chemical characteristic of CBD as a fatty acid exerts membrane disruption on bacterial species, particularly Gram-negative species [19,22]. While these studies have outlined the efficacy of CBD against several clinically relevant bacterial strains [17,18,19,22], there remain questions concerning the mechanisms of membrane disruption and the potential application of CBD as a therapeutic against *Salmonella* spp. Even though CBD has been shown to be effective in inducing bacterial death, little is known about the possibility of CBD-antibiotic co-therapy against bacterial infections. An interesting aspect of therapeutic design is the utilization of co-therapies whose efficacy is conferred through synergistic relationships between two or more therapeutic agents [30,31,32,33]. Determination of synergistic relationships between novel therapeutic agents such as CBD with other therapeutic agents could further advance the development of CBD as an innovative antibacterial. Considering this, our study seeks to explore the effectiveness of CBD-broad-spectrum antibiotic co-therapies in vitro against *S. typhimurium*, a clinically relevant Gram-negative bacterium.

This study utilized fluorescence microscopy, biological assays, and bacterial growth kinetics to determine the effectiveness of CBD-broad-spectrum antibiotic co-therapy against *S. typhimurium*. We hypothesize that CBD will exhibit a synergistic relationship with ampicillin, kanamycin, and polymyxin B against *S. typhimurium*. We expect that the membrane disruption carried out by CBD paired with the antibacterial mechanisms of these broad-spectrum antibiotics will induce more cell death in *S. typhimurium* and exhibit a synergetic relationship between these two agents. Previous studies from our group have explored the effectiveness of *C. sativa* CBD extracts against *Salmonella* species. In this study, we further explored the activity of this agent for co-therapy application with broad-spectrum antibiotics. Using several biological assays for synergy we observe interactions between CBD and three broad-spectrum antibiotics—ampicillin, kanamycin, and polymyxin B—which are three of the most prescribed antibiotics for *Salmonella* infections. The results of this study confirmed that CBD exhibits a synergetic relationship with ampicillin and polymyxin B against *S. typhimurium*. Extended kinetics also confirmed that co-therapy treatments did not result in developed resistance over the span of 24 h. The results of these studies provide new insight into the potential of CBD in co-therapy applications, thus progressing the research and development of CBD as a potential antibacterial agent.

## 2. Materials and Methods

### 2.1. Bacterial Strains

*Salmonella typhimurium* LT2 strain MS1868 (a gift from Dr. Anthony R. Poteete, University of Massachusetts) were utilized within this study. Luria Broth (BD, Difco, Franklin Lakes, NJ, USA) was used for bacterial cultures. Overnight cultures were created utilizing single colonies. Luria Broth cultures were incubated at 37 °C and 80 rpm shaking overnight. Plate Cultures were grown using LB agar (BD, Difco, Franklin Lakes, NJ, USA) through streaking from overnight stock culture and incubating at 37 °C for 6 h. Cultures screened regularly for avoidance of contamination.

### 2.2. CBD Extraction and Preparation

CBD extraction was carried out by Sustainable CBD LLC. (Salem, AL, USA). *C. sativa* biomass was weighed, tagged, and recorded in a receiving trailer for processing. Following storage, the biomass was reduced to between 200 and 500 microns and underwent CO_2_ extraction in an Apeks Transformer for subcritical extraction (Gibraltar Industries Inc., Buffalo, NY, USA). Subcritical extraction was carried out at a target pressure of 1200 psi, chiller temperature of 20–25 °C, propylene glycol percentage of 10%, orifice size of 22, resultant separator pressure of 350–400 psi, resultant separator temperature of −6–4 °C, for an extraction time of approximately 2–3 h. Following subcritical extraction, samples were prepared for decarboxylation prior to supercritical extraction. Hemp biomass was placed in the oven for approximately 100 min at 265 °C to decarboxylate. Once decarboxylation was carried out, an Apeks Transformer was utilized for supercritical extraction (Gibraltar Industries Inc., Buffalo, NY, USA). Supercritical extraction was carried out at a target pressure of 1800 psi, chiller temperature of 37–42 °C, propylene glycol percentage of 10%, orifice size of 18, resultant separator pressure of 350–400 psi, resultant separator temperature of 0–10 °C, for an extraction time of approximately 1–2 h per pound of material. The resulting material then underwent winterization through addition of ethanol to the crude extract. This sample was frozen and then filtered through Buchner funnels and the remaining ethanol was evaporated using a Heidolph rotary evaporator (Heidolph Instruments GmbH & Co. KG, Kelheim, Germany). Distillation was carried out using the Lab Society 5 L short path distillation unit (Lab Society^®^, Boulder, CO, USA) for further refinement. The resulting product of this procedure was winterized cannabinoid [40]. The winterized CBD crude oil obtained from Sustainable CBD LLC. was diluted with 4% EtOH and vortexed to a final concentration of 50 mg/mL. From this stock, serial dilutions were utilized to create our experimental concentrations of 1 μg/mL, 0.1 μg/mL, 0.01 μg/mL, and 0.001 μg/mL. CBD variety ‘Suver Haze’ was utilized for these studies.

### 2.3. Plate Screening for Antibacterial Activity

For preliminary confirmation of antibacterial activity of CBD, ampicillin, kanamycin, and polymyxin B against *S. typhimurium*, plate assays were utilized. Prior to the assays, CBD was diluted to concentrations of 1 μg/mL, 0.1 μg/mL, 0.01 μg/mL, and 0.001 μg/mL. Ampicillin (Sigma-Aldrich, Merck KGaA, Darmstadt, Germany) kanamycin (Sigma-Aldrich, Merck KGaA, Darmstadt, Germany), and polymyxin B (Sigma-Aldrich, Merck KGaA, Darmstadt, Germany) were diluted to concentrations of 50 μg/mL, 5 μg/mL, 0.5 μg/mL, and 0.05 μg/mL. *S. typhimurium* was incubated until the late log-phase (OD^600^ > 1) in sterile Luria broth. OD^600^ > 1 is representative of bacteria reaching the stationary phase of growth and the end of exponential growth [26].

To conduct the Kirby-Bauer assay, plated were inoculated with overnight cultures of *S. typhimurium* and top agar (BD Difco, Franklin Lakes, NJ, USA) to create a bacterial lawn. These plates were then divided into quarters representative of the four dilutions of CBD (1 μg/mL, 0.1 μg/mL, 0.01 μg/mL, and 0.001 μg/mL), ampicillin, kanamycin, or polymyxin B (50 μg/mL, 5 μg/mL, 0.5 μg/mL, and 0.05 μg/mL). Sterile paper disks were then soaked in a designated dilution of CBD, ampicillin, kanamycin, polymyxin B, or dH_2_O (control) and applied to the agar plates in triplicate. Resulting in three discs per CBD, ampicillin, kanamycin, or polymyxin B dilution on the agar plates. Plates were incubated at 37 °C for 24 h and the zones of inhibition (ZOI) were measured to quantify antibacterial activity. This assay was completed with biological replicates in triplicate [19,26].

For further confirmation of antibacterial activity, spot assays were conducted using lag-phase *S. typhimurium* cultures (OD^600^ > 1). Sterile plates were inoculated with four 10 μL aliquots of *S. typhimurium* culture. Each ‘dot’ was then inoculated with 10 μL of each CBD dilution (1 μg/mL, 0.1 μg/mL, 0.01 μg/mL, and 0.001 μg/mL) or 10 μL of antibiotic (ampicillin, kanamycin, or polymyxin B) dilution (50 μg/mL, 5 μg/mL, 0.5 μg/mL, and 0.05 μg/mL). Control ‘dots’ were treated with 10 μL of dH_2_O. Plates were then incubated for 24 h at 37 °C and observed for inhibition of bacterial growth. This assay was completed with biological replicates in triplicate [19,26].

### 2.4. CBD-Antibiotic Synergy Assay

To study the potential synergy between the three broad-spectrum antibiotics and CBD, the checkerboard assay was utilized [19,41]. 180 μL of *S. typhimurium* was seeded into a 96 well plate (Fisherbrand™, Fisher Scientific, Fair Lawn, NJ, USA) at an Optical Density measured at 600 nm (OD^600^) ≈ 0.15 (16.5 log_10_ CFU/mL). OD was standardized through all biological replicates. Ampicillin, kanamycin, and polymyxin B were serially diluted to concentrations of 50, 5, 0.5, and 0.05 μg/mL. CBD was serially diluted to concentrations of 1, 0.1, 0.01, and 0.001 μg/mL. *S. typhimurium* was then treated with 20 μL of CBD and 20 μL of antibiotic. This method allows for the potential synergy of ampicillin, kanamycin, and polymyxin B with CBD against *S. typhimurium* to be quantitavely examined at a wide variety of concentrations and combinations [41]. Synergistic effects were determined through OD^600^ measures from the checkerboard assay and fractional inhibitory concentration index (FICI) following 24 h of incubation at 37 °C. FICI is widely accepted and utilized for the quantitative determination of synergy between two agents [41]. The FICI was derived from minimum inhibitory concentrations (MIC) and calculated as below:
(1)FIC Antibiotic (^AB^) = MIC^AB^ in Combination/MIC^AB^ alone;(2)FIC Cannabidiol Extract (^CBD^) = MIC^CBD^ in Combination/MIC^CBD^ alone;(3)FICI = FIC^AB^ + FIC^CBD^


The FICI was evaluated as follows: synergy (FICI < 0.5), partial synergy (0.5 ≤ FICI ≥ 0.75), additive (0.76 ≤ FICI ≥ 1), indifference (1 ≤ FICI ≥ 4), or antagonism (FICI > 4). The average MIC from three biological replicates were used for these equations [41].

Percent inhibition calculations quantify reduction of bacterial growth 24 h following treatment utilizing control OD^600^ as the baseline value for inhibition. Percent inhibition was calculated as below:(*Eq*.)% Inhibition = 100% − ((Experimental^OD600^/Control^OD600^) × 100%)
Experimental^OD600^ = OD^600^ value of *S. typhimurium* treated with antibiotic monotherapy, CBD monotherapy, or CBD-antibiotic co-therapy.
Control^OD600^ = OD^600^ value of untreated *S. typhimurium*.

Percent Inhibition was used as a comparative measure of various treatments to assess synergistic activity. The average OD^600^ from three independent experiments were used for these equations. The average OD^600^ from three biological replicates were used for these equations [41].

### 2.5. Immunofluorescence Live-Dead Assay

To examine the antibacterial activity of CBD- antibiotic co-treatment, the live-dead assay was utilized with SYTO-9 (Invitrogen, Waltham, MA, USA) and propidium iodide (Sigma-Aldrich, St. Louis, MO, USA) following protocols published by Ayariga et al. [42]. *S. typhimurium* was grown overnight then diluted to an OD^600^ of 0.2 using LB broth. A dH_2_O treated and a 2% SDS treated *S. typhimurium* sample served as controls. Experimental samples consisted of antibiotic treated samples at concentrations of 0.5 μg/mL and 5 μg/mL, and CBD- antibiotic co-treatments with concentrations derived from synergetic potential in the checkerboard assay. Samples were incubated at 37 °C for 3 h. Afterwards, samples were stained with 1 X SYTO-9 and 40 μg/mL of propidium iodide and left at room temperature for 25 min, without exposure to light. Samples were then visualized using an EVOS FLC microscope (Life Technologies Corporation, Carlsbad, CA, USA).

### 2.6. Kinetic Studies

#### 2.6.1. Bacterial Growth Kinetics

*S. typhimurium* cells were freshly revived by sub-culturing on a Luria agar plate. To obtain fresh cultures for the experiments, an inoculum was transferred into LB and was grown overnight at 37 °C. To study bacterial growth kinetics, a 96-well plate was inoculated with 180 μL of overnight bacterial culture of *S. typhimurium* at a standardized OD^600^ ≈ 0.15 (16.5 log_10_ CFU/mL). The cultures were then incubated at 37 °C with rotary shaking at 121 rpm. Measurements of bacterial density (OD^600^) were taken every hour for 24 h using a spectrometer (Molecular Devices SpectraMax^®^ ABS Plus) (Molecular Devices LLC, San Jose, CA, USA). This experiment was completed with biological replicates in triplicate [19,25,43].

#### 2.6.2. Bacterial Growth Kinetics in the Presence of CBD

*S. typhimurium* cells were freshly revived by sub-culturing on a Luria agar plate. To obtain fresh cultures for the experiments, an inoculum was transferred into LB and was grown overnight at 37 °C. To study how CBD affects bacterial growth kinetics, three 96-well plates were inoculated with 180 μL of overnight bacterial cultures of *S. typhimurium* at an OD^600^ ≈ 0.15 (16.5 log_10_ CFU/mL). These cultures were then treated with 20 μL of varying concentrations of CBD (1, 0.1, 0.01, and 0.001 μg/mL). The cultures were then incubated at 37 °C with rotary shaking at 121 rpm. Measurements of bacterial density (OD^600^) were taken hourly for 24 h using a spectrometer (Molecular Devices SpectraMax^®^ ABS Plus). This experiment was completed with biological replicates in triplicate [19,25,43].

#### 2.6.3. Bacterial Growth Kinetics in the Presence of Antibiotics

*S. typhimurium* cells were freshly revived by sub-culturing on a Luria agar plate. To obtain fresh cultures for the experiments, an inoculum was transferred into LB and was grown overnight at 37 °C. To study how antibiotics affects bacterial growth kinetics, three 96-well plates were inoculated with 180 μL of overnight bacterial cultures of *S. typhimurium* at an OD^600^ ≈ 0.15 (16.5 log_10_ CFU/mL). These cultures were then treated with 20 μL of varying concentrations of ampicillin, kanamycin, or polymyxin B (50, 5, 0.5, and 0.05 μg/mL). The cultures were then incubated at 37 °C with rotary shaking at 121 rpm. Measurements of bacterial density (OD^600^) were taken hourly for 24 h using a spectrometer (Molecular Devices SpectraMax^®^ ABS Plus). This experiment was completed with biological replicates in triplicate [19,25,43].

#### 2.6.4. Comparative Bacterial Growth Kinetics between CBD and Antibiotic Mono-Treatment and CBD-Antibiotic Co-Treatment

To compare the effects of CBD and antibiotic mono-treatment with CBD-antibiotic co-treatment, a bacterial kinetic study was carried out utilizing co-treatment concentrations that exhibited synergy within the checkerboard assay. These synergistic concentrations included CBD (1 μg/mL)-ampicillin (5 μg/mL), CBD (0.1 μg/mL)-kanamycin (5 μg/mL), and CBD (1 μg/mL)-polymyxin B (0.5 μg/mL). *S. typhimurium* cells were freshly revived by sub-culturing on a Luria agar plate. To obtain fresh cultures for the experiments, an inoculum was transferred into LB and was grown overnight at 37 °C. 96-well plates were inoculated with 180 μL of overnight bacterial cultures of *S. typhimurium* at an OD^600^ ≈ 0.15 (16.5 log_10_ CFU/mL). Three 96-well plates were utilized per CBD-antibiotic co-treatment totaling nine plates. Cultures on each 96-well plate were treated with 20 μL of dH_2_O, antibiotic at the MIC concentration (consistent with antibiotic contained within co-treatment), CBD at the MIC concentration, or CBD-antibiotic co-treatment. The cultures were then incubated at 37 °C with rotary shaking at 121 rpm. Measurements of bacterial density (OD^600^) were taken hourly for 24 h using a spectrometer (Molecular Devices SpectraMax^®^ ABS Plus). This experiment was completed with biological replicates in triplicate [19,25,43].

### 2.7. Statistical Analysis

All experiments were performed on independent biological replicates. Statistical significance was determined for control and experimental groups using paired sample *t*-test. Data points were excluded if contamination was identified. Statistical analyses were preformed using Microsoft Excel (Microsoft 2010, Redmond, Washington, DC, USA). Values reported as mean ± SE (N = 3), *p*-values ≤ 0.05 were considered statistically significant.

## 3. Results and Discussion

### 3.1. Synergetic Characteristics of CBD-Broad-Spectrum Antibiotic Co-Treatment

To observe the potential synergistic characteristics of CBD-antibiotic co-therapy against *S. typhimurium*, the checkerboard assay and fluorescence microscopy was utilized [21]. The checkerboard assay is utilized to determine synergy between the broad-spectrum antibiotics and CBD. Through this synergy assay we were able to quantitatively determine synergistic activity between CBD and our three broad-spectrum antibiotics against *S. typhimurium* through measurement of optical density. Fluorescent imagery further demonstrates the antibacterial activity of CBD-antibiotic co-treatment and suggest that this treatment method results in greater antibacterial activity than that of antibiotic mono-treatment against *S. typhimurium*. Red fluorescence, signifying cell death, is seen as the dominant color within all co-treatments suggesting the successful antibacterial activity of these combinations against our target bacterium. In comparison to combination treatment, mono-treatment, emitted greater amounts of green fluorescence representative of viable living cells remaining within the culture. The results ultimately suggest that ampicllin and polymyxin B cotreatment with CBD presents significantly more antibacterial activity than observed in the mono-antibiotic treatments.

#### 3.1.1. Ampicillin-CBD Co-Treatment

In comparison to ampicillin mono-treatment, it was observed that an addition of CBD at a concentration of 1 μg/mL to ampicillin at concentrations of 50 μg/mL, 5 μg/mL, 0.5 μg/mL, or 0.05 μg/mL resulted in a greater inhibition of *S. typhimurium* growth (Figure 1A). At concentrations of 0.5 μg/mL ampicillin and 1 μg/mL CBD in co-treatment was observed as the treatment with the highest inhibitory effect determined through comparison to ampicillin mono-treatment at both the MIC concentration (5 μg/mL) and the experimental ampicillin concentration (0.5 μg/mL) (Figure 1A). CBD-ampicillin co-treatment resulted in a lower OD^600^ than that of ampicillin mono-treatment at the MIC concentration suggesting that co-treatment was more effective in reducing the bacterial growth over the 24 h period than the standard mono-treatment. Live/Dead staining results further confirmed the antibacterial activity of CBD-ampicillin co-treatment (Figure 1B). The abundance of red fluorescence paired with the ‘clumping’ effect as the result of cell lysis exhibits the antibacterial activity of the CBD-ampicillin co-treatment (Figure 1B). These results suggest that ampicillin and CBD exhibit a synergetic antibacterial interaction that leads to greater inhibition than ampicillin alone (Figure 1).

#### 3.1.2. Kanamycin-CBD Co-Treatment

In comparison to kanamycin mono-treatment, concentrations of 1 μg/mL CBD and 50 μg/mL of kanamycin in co-treatment resulted in a greater inhibition of *S. typhimurium* growth after 24 h (Figure 2A). These results suggest that CBD-kanamycin co-treatment produced a greater inhibitory effect than single treatments of kanamycin at any concentration. While the inhibitory effect was increased, the effect did not indicate synergy. Considering this, the combination can be most accurately described as indifferent. Activity between these two agents was limited at 50 μg/mL and 5 μg/mL kanamycin concentrations and there was no observed synergy at lower kanamycin concentrations (Figure 2A). Immunofluorescence staining results reached a similar conclusion with similar effectiveness between kanamycin at the MIC concentration and kanamycin-CBD co-treatment. These results in whole suggest that kanamycin effectiveness is not impacted by the addition of CBD in both the checkerboard assay as well as the staining assay (Figure 2B).

#### 3.1.3. Polymyxin B-CBD Co-Treatment

The final antibiotic that exhibited promising antibacterial synergy with CBD was polymyxin B. Results from both the checkerboard assay as well as Immunofluorescence staining suggest that these two agents exhibit a synergetic relationship (Figure 3). Results from the checkboard assay suggests a synergistic relationship at the concentration of 1 μg/mL CBD and 0.5 μg/mL Polymyxin B against *S. typhimurium* (Figure 3A). Polymyxin B mono-treatment at a MIC of 5 μg/mL exhibited a percent inhibition of 83%, whereas co-treatment with Polymyxin at 0.5 μg/mL and CBD at 1 μg/mL resulted in a percent inhibition of 92% (Figure 4). These results suggest that addition of CBD to polymyxin B can significantly reduce the required MIC while still retaining equivalent antibacterial activity. Immunofluorescence staining further confirmed the antibacterial activity that polymyxin B and CBD exhibit in treatment (Figure 3B). While co-treatment with polymyxin B and CBD resulted in a significant increase in inhibition, results from the checkerboard assay also exhibit potential antagonism at various concentrations. Antagonism can be observed in lower CBD concentrations at the MIC Polymyxin B concentration (Figure 3A). Results suggest that lower concentrations of CBD in combination with polymyxin B may partially inhibit the success of the antibiotic ability to kill the bacteria. This dose-dependent effect, best described as nonlinear mixture response, of these drugs on *S. typhimurium* places a critical emphasis on effective dosage [44]. Similar phenomenon has been observed with other co-therapies that feature synergetic and antagonistic effects in varied concentrations [45,46,47]. A potential disadvantage of this characteristic is the impact of multiple doses on the retention of synergetic activity. Examining the antibacterial effectiveness of this co-treatment in vivo would further define this phenomenon and progress our understanding of this potential co-treatment. While this antagonism is important to note, it does not discredit the synergy that polymyxin B and CBD possesses at other concentrations.

#### 3.1.4. Comparative Efficacy of CBD-Antibiotic Co-Treatment and CBD or Antibiotic Mono-Treatment

To further quantify the effectiveness of CBD- antibiotic co-treatment and to compare mono-treatment to co-treatment, percent inhibition was calculated from the checkerboard assay (Figure 4). Percent inhibition serves to comparatively assess OD^600^ between treated samples with untreated samples serving as the baseline. These measures further confirmed the synergy exhibited between both ampicillin and polymyxin B, with CBD. Additionally, this measure further suggests the indifferent activity between kanamycin and CBD. We observe significant increases in percent inhibition in both polymyxin B and ampicillin when in co-treatment with CBD, further suggesting the synergetic activity between these agents. It is observed that this co-treatment strategy significantly increases the effectiveness of CBD, polymyxin B, and ampicillin against *S. typhimurium* in comparison to mono-treatment.

#### 3.1.5. Synergistic Interaction Analysis

To further confirm the results obtained through checkerboard assay (Figure 1, Figure 2, Figure 3 and Figure 4), synergistic effects were further quantified through FICI. The FICI was calculated for the three combinations that presented potentially synergetic effects within the checkerboard assays (Figure 5, Table 1). The FICI results quantified what was observed through the checkerboard assays and qualified CBD-Ampicillin and CBD-Polymyxin B as having synergistic activity against *S. typhimurium*. FICI calculations determined that the relationship between CBD-Kanamycin could be described as indifference (Figure 5).

### 3.2. Comparative Kinetics of CBD-Antibiotic Co-Treatment

Broad-spectrum antibiotics have been widely used as a common treatment method against *Salmonella* spp. To compare the efficacy of antibiotic mono-treatment and CBD-antibiotic co-treatment, bacterial growth kinetics were recorded over the span of 24 h. These studies serve to further define and characterize the synergetic antibacterial activity that these CBD-antibiotic combinations possess against *S. typhimurium*. The impact on *S. typhimurium* growth was compared between antibiotic mono-treatment, CBD mono-treatment, and CBD-antibiotic co-treatment.

The three co-treatments examined were selected based on the synergistic activity observed in the checkerboard assays. The three co-treatments examined were ampicillin (0.5 μg/mL)-CBD (1 μg/mL) (Figure 6A), kanamycin (5 μg/mL)-CBD (1 μg/mL) (Figure 6B), and polymyxin B (0.5 μg/mL)-CBD (1 μg/mL) (Figure 6C). CBD-ampicillin and CBD-polymyxin B co-treatments reduced *S. typhimurium* by 3 log_10_ CFU/mL and 5 log_10_ CFU/mL, respectively in comparison to mono-treatment at MIC concentrations over the 24 h period. We observed the highest inhibitory effects in CBD-ampicillin and CBD-polymyxin B whereas, in kanamycin the effect of treatment is relatively similar between kanamycin-mono-treatment and CBD-kanamycin co-treatment.

The results of these comparative kinetic assays showed that (1) Co-treatment was more effective than the MIC mono-treatment application in ampicillin and polymyxin B and (2) Co-treatment resulted in little resistance development over a 24 h period. The ability of *S. typhimurium* to rapidly develop resistance has been a considerable concern to public health [26]. In previous studies it was shown that *S. typhimurium* was able to develop resistance against CBD treatment 24 h after treatment [19]. For this reason, it is significant to note that in CBD-antibiotic co-treatments, there was limited resistance development over the 24 h period, suggesting that co-treatment could potentially reduce the ability of *S. typhimurium* to develop resistance.

In these studies, we observe that antibiotic concentrations lower than the MIC remain effective in reducing bacterial growth when paired with CBD as a synergistic additive. This effectiveness was clearly depicted particularly in ampicillin (Figure 6A) and polymyxin B (Figure 6C) where we observed that co-treatment resulted in significantly lower OD^600^ 24 h after treatment in comparison to antibiotics at MIC values. CBD-ampicillin and CBD-polymyxin B co-treatment resulted in a 3 log_10_ CFU/mL and 5 log_10_ CFU/mL reduction, respectively. This was as expected, as we observed the most synergistic activity between polymyxin B and ampicillin, with CBD in checkerboard assays.

In kanamycin, we observed a negligible difference in OD^600^ other the 24 h period in comparison to kanamycin MIC treatment (Figure 6B). This result correlated with the lack of synergy observed within the checkerboard assay and calculated through the FICI (Figure 5). Kanamycin is an aminoglycoside, targeting the 30S ribosomal subunit to terminate protein synthesis to carry out its antibacterial effect. Results suggest that CBD exhibited synergy only with antibiotics, such as ampicillin and polymyxin B, whose antibacterial mechanisms target the cellular membrane directly. Considering this, the indifferent effect of CBD on kanamycin performance emphasizes the importance of mechanistic similarity on co-treatment design and efficacy.

In this era of antibiotic resistance, it is crucial to develop new antibacterial agents as well as develop alternative treatment methods against these resistant bacteria. The lack of antibiotic development in the 21st century has enhanced the need for researchers to repurpose antibiotics and determine treatment methods that help retain their antibacterial function. The use of agents extracted from botanical sources as synergetic additives has been an area of growing interest over recent history [16,18,38,44]. This study explored the potential synergistic activity between *C. sativa* L. CBD extract and three broad-spectrum antibiotics: ampicillin, kanamycin, and polymyxin B. The results of these studies suggest that CBD does exhibit synergetic activity with both ampicillin and polymyxin B against *S. typhimurium*, providing a potentially effective co-treatment method. Further studies are necessary to evaluate the cytotoxic effects, potential delivery mechanisms, and the effect of CBD treatment on *S. typhimurium* gene expression to further progress the development of this potential co-treatment method.

## 4. Conclusions

The decrease in antibiotic development over the 21st century has exacerbated the need for new antibacterial agents as well as new methodologies designed to retain the efficacy of current antibiotics [23,24,25,26]. CBD extract from *C. sativa* has been presented as a promising antibacterial agent with in vitro efficacy against several relevant bacterial pathogens including *Staphylococcus aureus*, *Streptococcus pneumoniae*, *Salmonella* spp. *Clostridium difficile*, *Neisseria* spp., *Moraxella catarrhalis*, and *Legionella pneumophila* [17,18,19]. This antibacterial activity achieved through membrane disruption of both Gram-positive and Gram-negative bacterial species presents CBD as a unique and particularly effective class of antibacterial agents [48,49]. As research continues to characterize and define the antibacterial activity of CBD, it is integral to examine interactions and potential synergy between CBD and other currently used therapeutics. Co-therapy has been utilized recently to combat resistant and persistent bacterial infections, considering this, we observed the interactions of CBD and broad-spectrum antibiotic co-treatments against *S. typhimurium* through in vitro screening for antibacterial activity and potential synergy. Through a biological assay for synergy, this study was able to determine that CBD exhibited synergetic activity with ampicillin and polymyxin B against *S. typhimurium*, whereas CBD-kanamycin co-treatment presented an indifferent effect on *S. typhimurium* growth in comparison to kanamycin monotreatment. Results from the synergy assay paired with calculations of final inhibitory concentration index allowed for quantitative qualification of the synergistic interactions between CBD-ampicillin and CBD-polymyxin B with both of their FICI values qualifying as synergy (FICI < 0.5) [41]. Fluorescence microscopy further confirmed the antibacterial effect of CBD-antibiotic co-treatments and allowed for qualitative comparison to antibiotic monotreatment. Bacterial kinetic studies allowed for determination of comparative efficacy between co-treatment and monotreatment. These studies revealed that CBD-ampicillin and CBD-polymyxin B co-treatment reduced bacteria by approximately 3 log_10_ CFU/mL and 5 log_10_ CFU/mL, respectively compared to CBD and antibiotic monotreatment. Kinetic studies additionally confirmed the indifference between CBD-kanamycin cotreatment and kanamycin monotreatment where there was no significant difference. The result of these studies exhibits the effectiveness of CBD-antibiotic cotreatments as well as provides new insight into the interactions between these two antibacterial agents. The synergetic action of CBD with ampicillin and polymyxin B, but not with kanamycin, suggests that synergy is reliant on the mechanism of action used by the antibiotic. Ampicillin and polymyxin B both induce bacterial death through mechanisms that degrade the cell membrane, similar to the hypothesized mechanism of CBD [14,19,50]. The shared target of the bacterial cell membrane may induce the synergistic effect of these two agents when used in cotreatment together. This study improves our knowledge of CBD and its interactions with broad-spectrum antibiotics and the potential effectiveness in vitro. Future studies should evaluate aspects of CBD and CBD-antibiotic combinations including cell cytotoxicity and delivery, that are related to further development and progression into in vivo testing models.

## Figures and Tables

**Figure 1 microorganisms-10-02360-f001:**
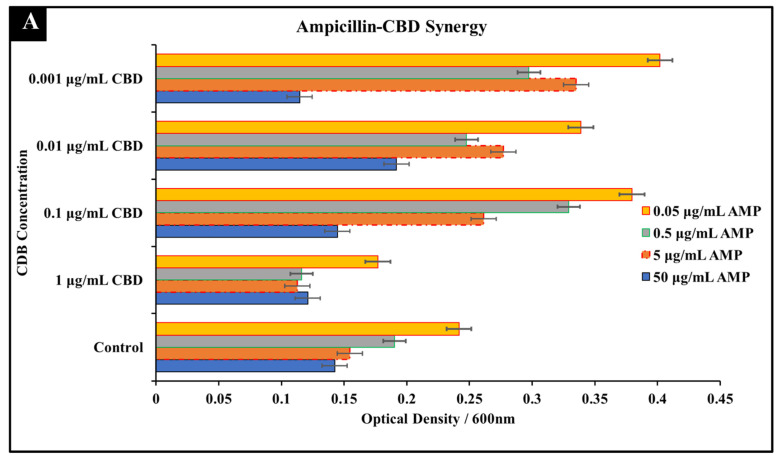
Synergistic analysis of ampicillin and CBD co-treatment of *S. typhimurium*. (**A**) Synergistic activity was observed at 0.5 μg/mL ampicillin and 1 μg/mL CBD co-treatment. Data presented as average OD^600^ ± SEM (**B**) Immunofluorescence staining analysis positive (2% SDS) and negative (dH_2_O) controls. Immunofluorescence staining of *S. typhimurium* treated with ampicillin at 0.5 and 5 μg/mL as well as the observed synergistic combination of 0.5 μg/mL ampicillin and 1 μg/mL CBD.

**Figure 2 microorganisms-10-02360-f002:**
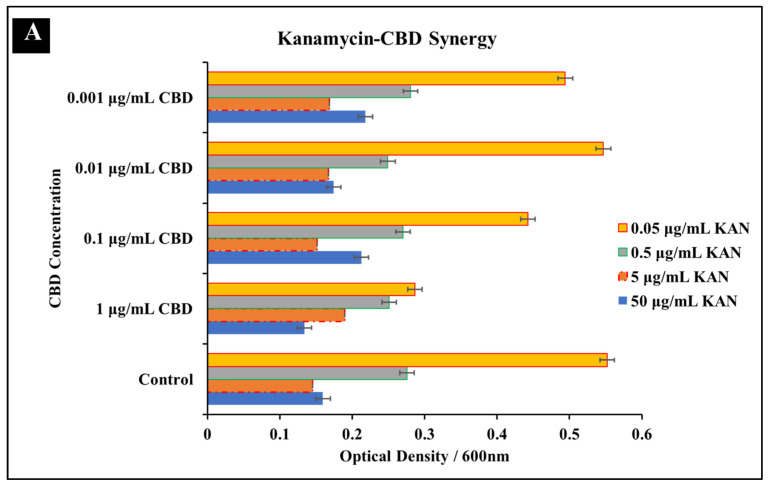
Synergistic analysis of kanamycin and CBD co-treatment of *S. typhimurium*. (**A**) Potential synergistic activity was observed at 5 μg/mL kanamycin and 1 μg/mL CBD co-treatment. Data presented as average OD^600^ ± SEM. (**B**) Immunofluorescence staining analysis positive (2% SDS) and negative (dH_2_O) controls. Immunofluorescence staining of *S. typhimurium* treated with kanamycin at 0.5 and 5 μg/mL as well as the observed synergistic combination of 5 μg/mL kanamycin and 1 μg/mL CBD.

**Figure 3 microorganisms-10-02360-f003:**
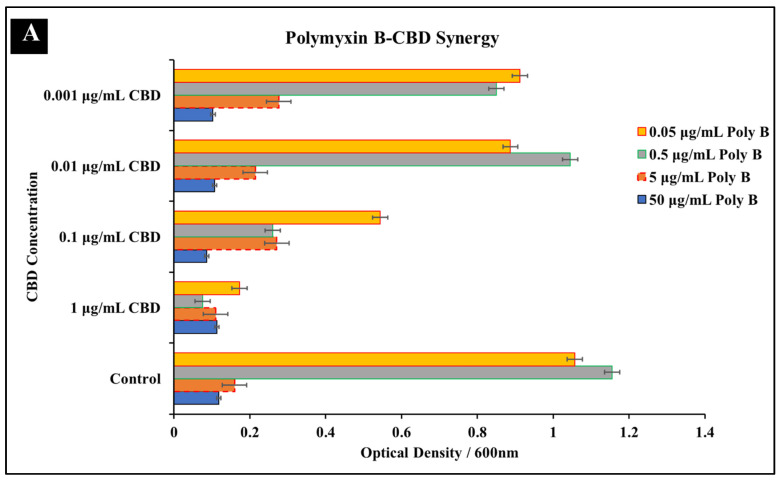
Synergistic analysis of polymyxin B and CBD co-treatment of *S. typhimurium*. (**A**) Potential synergistic activity was observed at 0.5 μg/mL polymyxin B and 1 μg/mL CBD co-treatment. Data presented as average OD^600^ ± SEM. (**B**) Immunofluorescence staining analysis positive (2% SDS) and negative (dH_2_O) controls. Immunofluorescence staining of *S. typhimurium* treated with polymyxin B at 0.5 and 5 μg/mL as well as the observed synergistic combination of 0.5 μg/mL polymyxin B and 1 μg/mL CBD.

**Figure 4 microorganisms-10-02360-f004:**
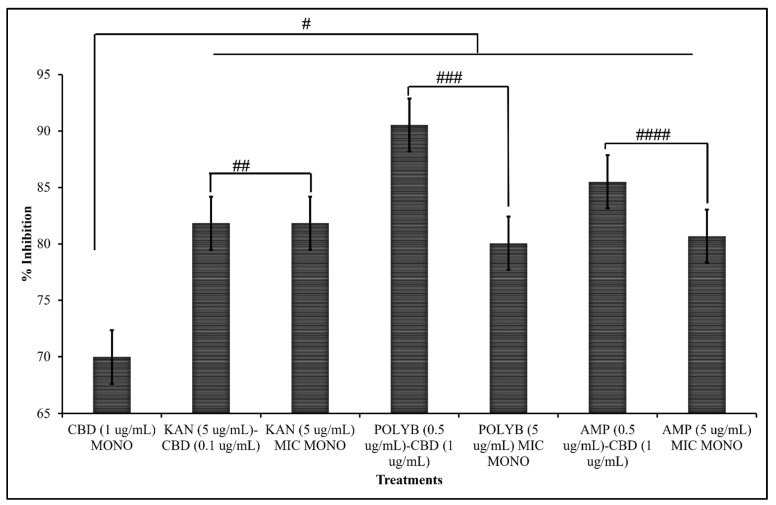
Percent Inhibition of both mono-treatment at MIC and co-treatment at synergistic concentrations as determined through checkerboard screening (SEM # = 0.059, ## = 0.018, ### = 0.006, #### = 0.062).

**Figure 5 microorganisms-10-02360-f005:**
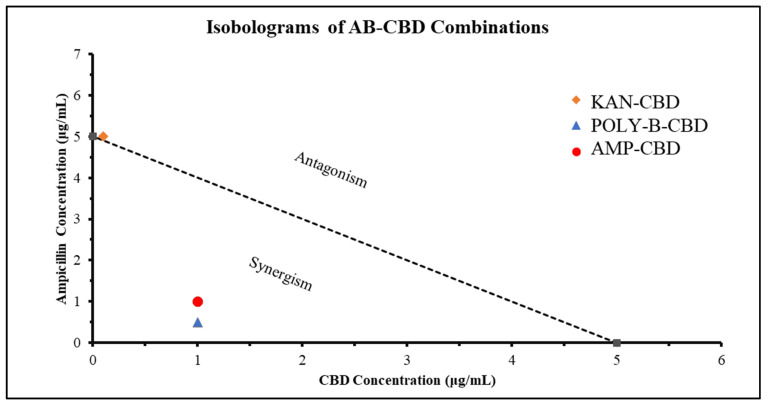
Synergistic activity of CBD and Antibiotics (Ampicillin (AMP), Kanamycin (KAN), Polymyxin-B (POLY-B)) against *S. typhimurium*. Isobologram showing the synergy (FICI < 0.5), partial synergy (0.5 ≤ FICI ≥ 0.75), additive (0.76 ≤ FICI ≥ 1), indifference (1 ≤ FICI ≥ 4), or antagonism (FICI > 4) effects of CBD paired with broad-spectrum antibiotics against *S. typhimurium*. Representative data for determination of FICI is included.

**Figure 6 microorganisms-10-02360-f006:**
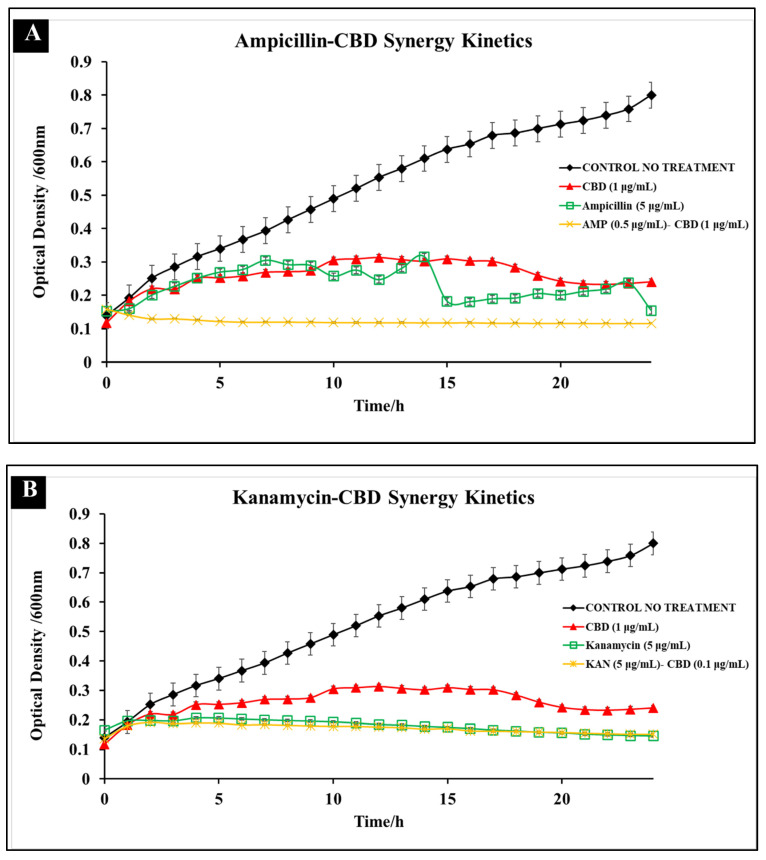
Comparative efficacy of CBD-antibiotic co-treatment and MIC antibiotic mono-treatment against *S. typhimurium*. Three co-treatments examined include ampicillin (0.5 μg/mL)-CBD (1 μg/mL) (**A**), kanamycin (5 μg/mL)-CBD (1 μg/mL) (**B**), and polymyxin B (0.5 μg/mL)-CBD (1 μg/mL) (**C**).

**Table 1 microorganisms-10-02360-t001:** Final Inhibitory Concentration Index (FICI) calculations for Ampicillin-CBD, Kanamycin-CBD, and Polymyxin B-CBD co-treatments. Effect of co-treatments described as synergy (FICI < 0.5), partial synergy (0.5 ≤ FICI ≥ 0.75), additive (0.76 ≤ FICI ≥ 1), indifference (1 ≤ FICI ≥ 4), or antagonism (FICI > 4).

	Antibiotic			Cannabidiol				
Combination	MIC Antibiotic Alone	MIC Combination	FIC Antibiotic	MIC CBD Alone	MIC Combination	FIC CBD	FICI	Effect
Ampicillin + CBD	5 μg/mL	0.5 μg/mL	0.1	5 μg/mL	1 μg/mL	0.2	0.3	Synergy
Kanamycin + CBD	5 μg/mL	5 μg/mL	1	5 μg/mL	0.1 μg/mL	0.02	1.02	Indifference
Polymyxin B + CBD	5 μg/mL	0.5 μg/mL	0.1	5 μg/mL	1 μg/mL	0.2	0.3	Synergy

## Data Availability

Data is contained within the article.

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
