# Peer review of "Cannabis sativa CBD Extract Exhibits Synergy with Broad-Spectrum Antibiotics against Salmonella enterica subsp. Enterica serovar typhimurium"

_microorganisms, 2022, doi:10.3390/microorganisms10122360_

Round 1
Reviewer 1 Report
In the manuscript entitled “Cannabis sativa CBD Extract Exhibits Synergy with Broad-Spectrum Antibiotics against Salmonella typhimurium,” Gildea and colleagues present a study to evaluate the effect of cannabidiol (CBD) effect on S. Typhimurium and their relationship with Broad-Spectrum Antibiotics against Salmonella Typhimurium. A study that describes the use of cannabidiol against Salmonella Typhimurium is very much worth to be published.
The study offers novel insights and provides new conclusions about this subject. The results showed that CBD has a synergetic relationship with ampicillin and polymyxin B against S. Typhimurium and additive or indifferent activity between kanamycin and CBD.
The authors are requested to use the correct nomenclature for Salmonella organisms: it must read: Salmonella enterica supsp. enterica serovar Typhimurium. (Please change the title of the manuscript).
Additional major weaknesses are listed below.
1. Please include section 3.1.1 in the numbering of the text.
2. Please organize the order of figure 1 according to what is stated in the text (lines 225 – 249).
3. Improve the text between lines 226 – 233. This information was described in materials and methods.
4. Place the letter A in figure 1 in the text.
5. Indicate in which part of the text figures 1A and 1B are described.
6. Indicate in which part of the text figures 1A and 1B are described.
7. Merge figures 1B and 1C.
8. Line 272: Please indicate figure 2A.
9. Merge figures 2B and 2C.
10. Line 288: indicate synergy or additive.
11. Line 287: Change the order of the phrase “Data presented as means…. in the figures legend1, 2, and 3.
12. Indicate in which part of the text figures 3A and 3B are described.
13. What are the disadvantages or risks of polymyxin having an antagonistic dose with CBD and synergistic with other doses? Please discuss. Has this phenomenon been observed in another paper?
14. Merge figures 3B and 3C.
15. It is necessary to make some subsections of the antibiotic experiments. Please avoid mixing the results of the antibiotics because they are confusing.
16. Please separate figure 5 and make a table of the FICI results
17. In the new table of the FICI results, please change the word “interpretation” to “effect” and indicate the FIC antibiotic and the FIC of the cannabidiol extract.
18. Please organize the order in which figure 6 is described in section 3.2.
19. Please reformulate the conclusion since it does not contribute to the document.
20. Regarding section 3.2. “Comparative Kinetics of CBD-antibiotic Co-treatment,” please include in the text, Information related to Salmonella concentration (Log 10 CFU/mL). These experimental data are essential to assess the credibility of the kinetic assay.
Author Response
Reviewer 1
In the manuscript entitled “Cannabis sativa CBD Extract Exhibits Synergy with Broad-Spectrum Antibiotics against Salmonella typhimurium,” Gildea and colleagues present a study to evaluate the effect of cannabidiol (CBD) effect on S. Typhimurium and their relationship with Broad-Spectrum Antibiotics against Salmonella Typhimurium. A study that describes the use of cannabidiol against Salmonella Typhimurium is very much worth to be published.
The study offers novel insights and provides new conclusions about this subject. The results showed that CBD has a synergetic relationship with ampicillin and polymyxin B against S. Typhimurium and additive or indifferent activity between kanamycin and CBD.
The authors are requested to use the correct nomenclature for Salmonella organisms: it must read: Salmonella enterica supsp. enterica serovar Typhimurium. (Please change the title of the manuscript).
Additional major weaknesses are listed below.
- Please include section 3.1.1 in the numbering of the text.
Section 3.1.1 is now included within text
- Please organize the order of figure 1 according to what is stated in the text (lines 225 – 249).
Text has been reorganized to properly state the contents of figure 1.
- Improve the text between lines 226 – 233. This information was described in materials and methods.
Text has been improved for clarity and relevance to results, not materials and methods.
- Place the letter A in figure 1 in the text.
Figure 1A is now present within text
- Indicate in which part of the text figures 1A and 1B are described.
Text now differentiates between figures 1A and 1B
- Indicate in which part of the text figures 1A and 1B are described.
Text now differentiates between figures 1A and 1B
- Merge figures 1B and 1C.
Figures are now merged
- Line 272: Please indicate figure 2A.
Figure 2A is now indicated.
- Merge figures 2B and 2C.
Figures are now merged.
- Line 288: indicate synergy or additive.
Synergy is now indicated.
- Line 287: Change the order of the phrase “Data presented as means…. in the figures legend1, 2, and 3.
Altered to: Data presented as average OD600 ± SEM
- Indicate in which part of the text figures 3A and 3B are described.
Text now differentiates between figures 3A and 3B.
- What are the disadvantages or risks of polymyxin having an antagonistic dose with CBD and synergistic with other doses? Please discuss. Has this phenomenon been observed in another paper?
Disadvantages of antagonistic doses now discussed and reference on antagonism in co-treatment now included.
- Merge figures 3B and 3C.
Figures are now merged.
- It is necessary to make some subsections of the antibiotic experiments. Please avoid mixing the results of the antibiotics because they are confusing.
Subsections now differentiate each antibiotic experiment within the text.
- Please separate figure 5 and make a table of the FICI results
Figure 5 now separated into figure 5 and table 1.
- In the new table of the FICI results, please change the word “interpretation” to “effect” and indicate the FIC antibiotic and the FIC of the cannabidiol extract.
Word changes to effect. MIC values included for CBD and antibiotics in monotreatment. FIC values for CBD and antibiotic are now added.
- Please organize the order in which figure 6 is described in section 3.2.
Section 3.2 reorganized to properly discuss figure 6 clearly
- Please reformulate the conclusion since it does not contribute to the document.
Conclusion rewritten to contribute to the document.
- Regarding section 3.2. “Comparative Kinetics of CBD-antibiotic Co-treatment,” please include in the text, Information related to Salmonella concentration (Log 10 CFU/mL). These experimental data are essential to assess the credibility of the kinetic assay.
Information regarding salmonella concentrations has been added throughout the manuscript.

Reviewer 2 Report
The work is better to new novel broad-spectrum CBD-antibiotic against Salmonella typhimurium. There is a question about the legal status of the Cannabis sativa, C. sativa is a addictive plant. The dose is big question. How to get the dose of CBD-antibiotic for world-wide use?
There are some minor comments:
In title, the Cannabis sativa should be italic
Author Response
Reviewer 2
The work is better to new novel broad-spectrum CBD-antibiotic against Salmonella typhimurium. There is a question about the legal status of the Cannabis sativa, C. sativa is an addictive plant. The dose is big question. How to get the dose of CBD-antibiotic for world-wide use?
There are some minor comments:
In title, the Cannabis sativa should be italic
Cannabis sativa is now italicized.

Reviewer 3 Report
Comments and suggestions:
- Please, revise the manuscript in view of using italic letters for all Latin names - from Title to Conclusion
- Rewrite sentence in lines 19 and 20 avoiding "here" and "we"
-please, check if is it correct to write Salmonella typhimurium or Salmonella Typhimurium?
- In the Introduction part, please provide a background of antimicrobial activity - chemical composition of CBD, potential chemical carrier of antimicrobial capacity, etc.
- Add state-of-the-art for antimicrobial resistance of Salmonella in view of the application of natural oils, extracts, and similar formulations
- Avoid using "we", put all sentences in a passive form.
- 2.1 Bacterial Strains - add incubation conditions in details, as well as preparation and manipulation steps for further experiments
- 2.3 Plate Screening for Antibacterial Activity - provide references and all other details to enable reproduction of methods; especially highlight what you mean when you wrote that OD600 value for Salmonella strains was greater than 1?
-Provide information on all used antibiotics including manufacturer and other
- rewrite sentences in lines 129 and 130
- try to present the method more simply 2.4 CBD-Antibiotic Synergy Assay
- I suppose that 2.6.2 Bacterial Growth Kinetics in the Presence of CBD and 2.6.3 Bacterial Growth Kinetics in the Presence of Antibiotics present some kind of Time kill kinetics study, so please add references where this methodology was used for Salmonella species.
- Conclusions lack all necessary information about gained results. I suggest writing much better paragraphs for highlighting the significance of the study.
Author Response
Reviewer 3
- Please, revise the manuscript in view of using italic letters for all Latin names - from Title to Conclusion
Manuscript revised and italic letters are now used for all Latin names.
- Rewrite sentence in lines 19 and 20 avoiding "here" and "we"
Sentence rewritten to avoid here and we.
-please, check if is it correct to write Salmonella typhimurium or Salmonella Typhimurium?
Salmonella typhimurium has been deemed as acceptable nomenclature by the NIH and CDC.
- In the Introduction part, please provide a background of antimicrobial activity - chemical composition of CBD, potential chemical carrier of antimicrobial capacity, etc.
Background of antimicrobial activity now included.
- Add state-of-the-art for antimicrobial resistance of Salmonella in view of the application of natural oils, extracts, and similar formulations
State of the art antimicrobial resistance of Salmonella now discussed within the realm of the manuscript.
- Avoid using "we", put all sentences in a passive form.
Corrections made throughout paper to avoid we and sentences placed in passive form.
- 2.1 Bacterial Strains - add incubation conditions in details, as well as preparation and manipulation steps for further experiments
Incubation conditions and details added as well as preparation and manipulation steps for further experiments.
- 2.3 Plate Screening for Antibacterial Activity - provide references and all other details to enable reproduction of methods; especially highlight what you mean when you wrote that OD600 value for Salmonella strains was greater than 1?
Reference now provided for methods. Meaning of OD600 > 1 now described within methods.
-Provide information on all used antibiotics including manufacturer and other
Information and manufacturer now provided.
- rewrite sentences in lines 129 and 130
Sentence rewritten to provide clarity.
- try to present the method more simply 2.4 CBD-Antibiotic Synergy Assay
Methods altered slightly to provide further specificity and clearly describe the methods used.
- I suppose that 2.6.2 Bacterial Growth Kinetics in the Presence of CBD and 2.6.3 Bacterial Growth Kinetics in the Presence of Antibiotics present some kind of Time kill kinetics study, so please add references where this methodology was used for Salmonella species.
References added for bacterial growth kinetic studies.
- Conclusions lack all necessary information about gained results. I suggest writing much better paragraphs for highlighting the significance of the study.
Conclusion rewritten to contribute to the document and describe significance.

Reviewer 4 Report
Dear editorial board
Greetings,
I pleased to read and comment on submitted paper “Cannabis sativa CBD Extract Exhibits Synergy with Broad Spectrum Antibiotics against Salmonella typhimurium” to the journal “microorganism”.
I have read it and following suggestions are now available to improve the paper before final decision,
1- A minor correction on the English, mainly as a polish seems necessary.
2- A complete background for application of cannabidiol should be inserted in introduction.
3- We need a response for this combination “CBD-antibiotic” whether it can be used in vitro?
4- Can we also recommend this combination for human purposes? We need some trials before we can conclude like this, I did not see such analysis in this version. Please clarify.
5- I think only a standard strain “Salmonella typhimurium LT2 strain MS1868” and the expected results are not satisfactory for a conclusion, we need some clinical data supporting the findings.
6- How the authors measured the cytotoxicity of this combination of CBD-antibiotic? MTT ASSAY? FLUCY? WHAT ?
7- The section of bacterial growth kinetics needs reference for its protocols.
Author Response
Reviewer 4
I pleased to read and comment on submitted paper “Cannabis sativa CBD Extract Exhibits Synergy with Broad Spectrum Antibiotics against Salmonella typhimurium” to the journal “microorganism”.
I have read it and following suggestions are now available to improve the paper before final decision,
- A minor correction on the English, mainly as a polish seems necessary.
English has been polished.
- A complete background for application of cannabidiol should be inserted in introduction.
Background of CBD application now included within introduction.
- We need a response for this combination “CBD-antibiotic” whether it can be used in vitro?
Yes, this manuscript shows in vitro efficacy of CBD-antibiotic combinations against S. typhimurium. The results and conclusions now further emphasize this.
- Can we also recommend this combination for human purposes? We need some trials before we can conclude like this, I did not see such analysis in this version. Please clarify.
Future studies will aim to further characterize and progress CBD into in vivo models. This study seeks to explore the concept and examine in vitro efficacy.
- I think only a standard strain “Salmonella typhimurium LT2 strain MS1868” and the expected results are not satisfactory for a conclusion, we need some clinical data supporting the findings.
Conclusion has been rewritten to further express the significance of this study. There is not any current clinical data on CBD-antibiotic combinations as this novel application has just begun to gain research interest.
- How the authors measured the cytotoxicity of this combination of CBD-antibiotic? MTT ASSAY? FLUCY? WHAT ?
This study did not explore cytotoxicity. Future studies will study cytotoxicity and in vivo efficacy.
- The section of bacterial growth kinetics needs reference for its protocols.
Section now includes references for protocols.

Round 2
Reviewer 1 Report
Please summarize the conclusions, answering what the results indicate, what new evidence your results provide and if is neccesary to prove key points with new studies.
Reviewer 2 Report
the version is better, there are no comments.
Reviewer 3 Report
Thank you for the acceptance all suggestions and improve your manuscript.